Optimization of extraction conditions and determination of purine content in marine fish during boiling

Li Tingting 1
Ren Likun 2
Wang Dangfeng 2
Song Minjie 2
Li Qiuying 2
Li Jianrong 2 lijr6491@163.com
1 College of Life Science, Dalian Minzu University , Dalian , China
2 College of Food Science and Technology, Bohai University , Jinzhou , China
Farag Mohamed
Electronic publication date: 2019 May 6
Publication date: 2019
Volume: 7
Electronic Location ID: e6690
Received 2018 Dec 7; Accepted 2019 Feb 27
Copyright: © 2019 Li et al.
Copyright year: 2019
Copyright holder: Li et al.
License: This is an open access article distributed under the terms of the Creative Commons Attribution License, which permits unrestricted use, distribution, reproduction and adaptation in any medium and for any purpose provided that it is properly attributed. For attribution, the original author(s), title, publication source (PeerJ) and either DOI or URL of the article must be cited.
License URL: https://creativecommons.org/licenses/by/4.0/

Keywords: Marine fish, Optimization, Purine extraction method, HPLC, Purine content, Boiling

Funding: National Natural Science Foundation of China 31471639 National Key R&D Program of China 2018YFD0400600, 2017YFD0400106 The Opening Foundation of University-Enterprise Alliance of Food Industry in Liaoning Province 2018LNSPLM0104 This work was supported by the National Natural Science Foundation of China (No. 31471639), the National Key R&D Program of China (No. 2018YFD0400600, 2017YFD0400106). The Opening Foundation of University-Enterprise Alliance of Food Industry in Liaoning Province (2018LNSPLM0104) provide financial support for supplementing related experiments during revision. The funders had no role in study design, data collection and analysis, decision to publish, or preparation of the manuscript.

==============================
Background

Gout is the second most common metabolic disease affecting human health. The disease of gout is closely related to the level of uric acid, which is the end-product of human purine metabolism. Moreover, food is the main way of external ingestion of purine.

Method

A simple and time-saving method was developed to extract purines like adenine, hypoxanthine, guanine, and xanthine from marine fish by single factor design combined with Box–Behnken. The contents of these purines in the edible parts and internal organs of marine fish, as well as Scophthalmus maximus, were determined by high-performance liquid chromatography to investigate the relationship between the boiling process and purine content.

Result

The mixed-acid method was chosen for the extraction of purine bases and the extraction conditions were as follows: mixture acid 90.00% TFA/80.00% FA (v/v, 1:1); hydrolysis temperature 90.00 °C; time 10.00 min; liquid-to-solid ratio 30:1. The total purine content of the edible parts (eyes, dorsal muscles, abdominal muscles, and skin) was the highest in Scophthalmus maximus, followed by sphyraena, Sardinella, Trichiurus lepturus, Scomberomorus niphonius, Pleuronectiformes, Sea catfish, Anguillidae, and Rajiformes. Moreover, boiling significantly reduced the purine content in the marine fish because of the transfer of the purines to the cooking liquid during boiling. Scophthalmus maximus, Sphyraena, and Sardinella were regard as high-purine marine fish, which we should eat less. We also confirmed that boiling significantly transferred purine bases from fish to cooking liquid. Thus, boiling could reduce the purine content of fish, thereby reducing the risk of hyperuricemia and gout.

Introduction

Gout is a type of inflammatory arthritis caused by the deposition of monosodium crystals in tissues. It has become the second-most common metabolic disease affecting human health (Goldberg et al., 2017). According to statistics, about 75 million people in China now suffer from gout or hyperuricemia. These diseases are strongly influenced by uric acid, which is a metabolite of the purine components of RNA and DNA present in foods. Normal serum urate levels in healthy adults are below 0.45 mmol/L for men and below 0.36 mmol/L for women (Zhao et al., 2005). However, when the blood uric acid exceeds normal levels, there is an increased risk for diseases such as gout.

Diet is a key factor in high serum urate levels, and some studies have confirmed that a low-purine diet significantly helps patients with hyperuricemia and gout (Lou, Lin & Benkmann, 2001; Shmerling, 2012). The consumption of food with high purine content can disrupt the balance between uric acid synthesis and metabolism, leading to hyperuricemia or eventual gout (Gowda et al., 2010). Therefore, adherence to a low-purine diet plays an important role in reducing serum uric acid concentration in humans (Suresh & Das, 2012). Compared with other meat products, fish is usually labeled as a healthier food because of its high nutritional value (e.g., high levels of unsaturated fatty acids). However, this nutritional advice may mislead consumers who are not aware of the intrinsic purine content of fish. The quantity and types of purine bases (adenine, guanine, hypoxanthine, and xanthine) in food, however, might be altered by different cooking methods (Brulé, Sarwar & Savoie, 1989). Previous studies have shown that various processing methods can reduce the purine contents in foodstuff. Lou, Lin & Benkmann (2001) evaluated the effects of processing on reducing purines contents in tilapia, with three cooking treatments (boiling, steaming, and microwave cooking) were carried out. The results showed that the content of adenine and hypoxanthine decreased 46.40% during boiling, which achieved the highest ratio of purine removal. And the effects of cooking on purine concentration in beef and chicken has also been widely studied, but little information is available on the purine content in the edible parts of other marine fish. There are many methods for simultaneous quantification of the purine levels in food. Klampfl et al. (2002) analyzed the purine and pyrimidine contents in beer by capillary zone electrophoresis. Similarly, an ultra-high-performance liquid chromatography–tandem mass spectrometry (UPLC–MS/MS) method was used to analyze seven purines and pyrimidines in pork products (Clariana et al., 2010). High-performance liquid chromatography (HPLC) has become the most widely used method for purine detection because of its high efficiency, convenience, and accuracy (Rong et al., 2015). Other studies have investigated the purine contents of vegetarian meat analogues, beer, beer-like alcoholic beverages, pork, and beef via HPLC (Havlik et al., 2010; Fukuuchi et al., 2013; Rong et al., 2015). Nevertheless, few HPLC methods have been used to quantify the adenine, guanine, hypoxanthine, and xanthine contents in marine fish (Qu et al., 2017).

In this paper, a simple and reliable method for purine extraction is reported. Furthermore, the purine contents (adenine, guanine, hypoxanthine, and xanthine) in the edible parts of nine marine fish were measured by HPLC. Finally, the effect of boiling on the purine content in the edible parts of marine fish was investigated in the context of its viability in decreasing the risk of gout attacks caused by diet.

Material and Methods

Chemicals and reagents

Purine standards (adenine, guanine, hypoxanthine, and xanthine) were purchased from Shanghai Aladdin Bio-Chem Technology Co., Ltd (Shanghai, China). The standards were all chromatography-grade with purity >98%. Chromatography-grade glacial acetic acid, methyl alcohol, tetrabutylammonium hydroxide, and trifluoroacetic acid (TFA) were also obtained from Shanghai Aladdin Bio-Chem Technology Co., Ltd. Analytically pure formic acid (FA) and perchloric acid (PCA) were purchased from Tianjin Fengchuan Chemical Reagent Co., Ltd (Tianjin, China). Water was purified by a Milli-Q system (Millipore, Burlington, MA, USA).

Sampling and pretreatment

Sample collection

Live marine fish (Scophthalmus maximus, Scomberomorus niphonius, Trichiurus lepturus, Pleuronectiformes, Sea catfish, Sardinella, Sphyraena, Anguillidae, and Rajiformes) were purchased from the local wholesale seafood market in Jinzhou, China, transported to the laboratory where they were killed, and then the dorsal muscles, abdominal muscles, skin, eyes, and viscera were removed for testing. All samples were minced and stored at 0 °C before proceeding.

Sample pretreatment

The edible parts of the marine fish (dorsal muscles, abdominal muscles, and skin) were separately boiled in water for 3, 6, 9, 12, and 15 min, and the contents of the purine bases in all samples (edible parts and cooking liquids) were determined. The uncooked samples were used as a control group. Three independent measurements were taken, and the mean and standard deviations were calculated.

Establishment of purine extraction method

The purine bases in the samples were extracted according to the method of Piñeiro-Sotelo, López-Hernández & Simal-Lozano (2002), with some modifications. First, 200 mg of sample was added to a centrifuge tube (50 mL) with 10 mL of acid, and heated at 90 °C in a water bath for 15 min. Next, the acid hydrolysate was placed in a rotary evaporator at 75 °C to remove the volatiles, and then redissolved in 10 mL of the HPLC mobile phase (water-methanol-glacial acetic acid-20% tetrabutylammonium hydroxide (v/v = 879/100/15/6). Finally, the sample was centrifuged at 8,000g for 10 min at 4 °C and filtered through a 0.22-μm filter before analysis by HPLC.

Single-factor design

PCA method

The optimum conditions for the purine bases extraction by the PCA method were determined by the single factor method and Box–Behnken design (BBD). The hydrolysis was performed following the procedure reported by Piñeiro-Sotelo, López-Hernández & Simal-Lozano (2002), with some modifications. A single factor method was employed in this study. Four factors were investigated which include PCA concentrations (5%, 10%, 20%, 30%, 40%, 50%, 60%, 70%, 80%, 90%, and 100%), the temperature of water bath (30, 40, 50, 60, 70, 80, and 90 °C), hydrolysis time (35, 40, 45, 50, 55, 60, 65, and 70 min), and liquid-solid ratio (10:1, 20:1, 30:1, 40:1, 50:1, 60:1, 70:1, 80:1, 90:1). When one factor was studies, others were fixed in the optimum value determined in this paper.

Mixed-acid method

The mixed-acid method was optimized in a similar manner as the PCA method. Single- factor experiments were performed to examine the influence of temperature in water bath (30, 40, 50, 60, 70, 80, 90, and 100 °C), Liquid-solid (10:1, 20:1, 30:1, 40:1, 50:1, 60:1, 70:1, 80:1, 90:1), the concentration of TFA and FA (45%, 50%, 55%, 60%, 65%, 70%, 75%, 80%, 85%, 90%, and 95%), and hydrolysis time (5, 10, 15, 20, 25, 30, 35, 40, 45, 50, 55, and 60 min) in extraction efficiency. The effect of each factor was explored by changing the factor while keeping other factors constant.

Plackett–Burman design for screening factors

In this study, five independent factors as hydrolysis temperature, hydrolysis time, FA concentration, TFA concentration, and liquid to solid ratio were selected to investigate the effective factors of the high extraction yield of purine. According to the result of single-factor test, each variable was studied at two levels (Farrokhnia et al., 2016): low (−1) and high (+1) which were selected from the maximum response interval of each independent variable (Table 1). A total of 12 experimental runs were designed by the software of Design Expert-8.0.6 (Stat-Ease, Inc., Minneapolis, MN, USA). The results of PBD were analyzed to select the most important factors, which were further optimized using BBD.

Table 1 Experimental variables and their levels used in the Plackett–Burman design of mix-acid method.

Variables	Levels	
Factor codes	Factor	Units	Low (−1)	High (+1)	
A	The temperature of hydrolysis	°C	80	100	
B	Liquid-solid ratio	ml/g	20	40	
C	The concentration of FA	%	75	85	
D	The concentration of TFA	%	80	90	
E	The time of sample hydrolysis	min	5	15	

Box–Behnken design for optimizing of purine extraction method

In this study, BBD was used to find the optimum extraction conditions for obtaining the highest extraction yield of total purine. Based on single-factor design and PBD, level ranges of factors and critical factors were determined and selected for response surface methodological analysis. Each of these variables was studied at three different levels (−1, 0, 1) (Grosso et al., 2014). Factors and levels of response surface methodology were shown in Table 2. The software Design Expert 8.0.6 was used for experimental design, data analysis and model building. Three-dimensional response surface plots and contour were drawn to identify the interaction between factors and responses.

Table 2 Levels of independent variables for Box–Behnken design experiment.

Variables	Levels	
Factor codes	Name	Units	Low (−1)	(0)	High (+1)	
(a) Experimental variables and their levels used in the BBD for PCA method.	
A	The concentration of PCA	%	70	80	90	
B	Hydrolysis temperature	°C	70	80	90	
C	The time of sample hydrolysis	min	50	55	60	
D	Liquid-solid ratio	ml/g	50	60	70	
(b) Experimental variables and their levels used in the BBD for mix-acid method.	
A	Hydrolysis temperature	°C	80	90	100	
B	The concentration of TFA	%	80	85	90	
C	The time of sample hydrolysis	min	5	10	15	

Validation test

Verification experiments were performed at the predicted conditions with some modified, indicating the validity of the predicted models. When the relative error between the predicted value and the actual value was less than 5%, it showed that the regression equation fits well with the actual situation and is accurate and reliable.

HPLC conditions

A Shimadzu LC-2030 HPLC system (Shimadzu Corporation, Kyoto, Japan) consisting of an LC-20AD pump unit, an SPD-20AV UV detector, and a CTO-20AC column heater, was used to identify the purine bases extracted. An Agilent Eclipse XDB-C18 column (4.6 mm × 250.0 mm × 5.0 μm; Agilent Technologies, Germany) was used as the analytical column and maintained at 28 °C during operation. The mobile phase was water-methanol-glacial acetic acid-20% tetrabutylammonium hydroxide (v/v = 879/100/15/6, pH = 3.44), and the 10 μL sample was eluted at a flow rate of 0.8 mL/min. At the end of each procedure, the analytical column was washed with the mobile phase for 30 min and equilibrated before the next run. The detector measured absorbance at 254 nm and data was analyzed using Shimadzu analysis software (Shimadzu Corporation, Kyoto, Japan).

Method evaluation

The purine base standards (adenine, guanine, hypoxanthine, and xanthine) were dissolved in ultrapure water at concentrations of 0.1, 0.5, 1, 5, 10, 50, 100, 200, and 300 mg/L and detected by HPLC according to the method described above. Purine bases in the samples were identified by comparing the peak retention times with those of the standard solutions. Quantification of the purine bases was based on the regression analysis of peak area against concentration. The linearity, range, squared correlation coefficient values (R2), and limits of detection (LODs) were determined (Li et al., 2015). The LOD was calculated as the concentrations corresponding to three times the S/N (Peng et al., 2008). The precision of the method and the repeatability of the purine extraction process were evaluated by testing the mixed purine base standard solution and the extraction sample from Scophthalmus maximus (dorsal muscles) six times, and the relative standard deviation (RSD%) was calculated. Recoveries of adenine, guanine, hypoxanthine, and xanthine were determined using samples in which the content of four kinds of purines had been determined. In each case known quantities of individual purine base standards with 0.5, 1, 2 times of the quantified contents was added into sample, which was then preformed to the optimum purine extraction method and then analyzed by HPLC (Havlik et al., 2010). The percent recovery was calculated according to the following formula (Peng et al., 2008):%Recovery=A−BC×100

Where A is the value of purine content of the sample that added purine base standards, B is the content of purine in the sample without purine base standards and C is the known content of purine bases standards that added in sample.

Statistical analysis

All measurements were performed in triplicate and the standard deviation was determined using SPSS 20 for Windows (SPSS Inc., Chicago, IL, USA). Graphs were drawn with the OriginPro 8.5 software package (OriginLab, Northampton, MA, USA). Statistical significance was assessed via analysis of variance (ANOVA). Differences with P < 0.05 were considered statistically significant.

Results and Discussion

HPLC method

The HPLC method described in this paper were used to simultaneously quantify the adenine, guanine, hypoxanthine, and xanthine contents in marine fish. As shown in Fig. 1, the four purine bases were separated within 10 min.

Figure 1 HPLC chromatogram of standards’ solution.

A, Adenine; B, Guanine; C, Hypoxanthine; D, Xanthine.

Establishment of the purine extraction method

Single factor design

PCA method

The results showed that acid concentration, hydrolysis temperature, hydrolysis time, and the liquid-solid ratio had influences on the extraction percent of the purines. As shown in Fig. 2, the extraction percent of the purine decreased gradually with increasing PCA concentration, eventually beginning to increase once the PCA concentration reached 60%. When the concentration of PCA was 80%, the extraction percent was higher. The extraction percent also reached a maximum at a hydrolysis temperature of 80 °C, beyond this point it decreased slightly. This may be due to the destruction of some purine bases at high temperatures (Jamil, Halim & Sarbon, 2016). In addition, the extraction with the PCA method was the best with a hydrolysis time of 55 min and a liquid-solid ratio of 60:1. Finally, level ranges of acid concentrations (70–90%), temperatures (70–90 °C), hydrolysis time (50–60 min) and liquid-solid ratio (50:1–70:1 mL/g) were selected for BBD.

Figure 2 Influence of different factors on the extraction percent using the PCA method.

(A) PCA concentration, (B) hydrolysis temperature, (C) hydrolysis time, and (D) liquid-solid ratio.

Mixed-acid method

The effects of hydrolysis temperature, the liquid-solid ratio, FA concentration, TFA concentration, and hydrolysis time on the extraction percent of the four purine bases are shown in Fig. 3. An increase in hydrolysis temperature improved the extraction efficiency gradually, until a maximum extraction percent was reached at about 90 °C. The extraction percent of the purine increased firstly and then decreased with the increase of TFA concentration, with the extraction percent reached the maximum at 85%. The extraction percent of the purines fluctuated with increasing liquid-solid ratio, and the highest extraction percent was obtained when the sample was hydrolyzed for 10 min. Moreover, the changes of FA concentration had little effect on the extraction percent of purine. Finally, according to the results of single factors design, the selected low and high values for hydrolysis temperature, liquid-solid ratio, FA concentration, TFA concentration, and hydrolysis time were (80, 100 °C), (20:1, 40:1 mL/g), (75%, 85%), (80%, 90%), and (5, 15 min) in the PBD, respectively.

Figure 3 Influence of different factors on the extraction percent using the mix-acid method.

(A) Hydrolysis temperature, (B) liquid-solid ratio, (C) FA concentration, (D) TFA concentration, and (E) hydrolysis time.

Plackett–Burman screening studies

The workload of a five factors response surface design was large, so its variables need to be screened. PBD was an effective method for screening the best candidate factors (Vasiee et al., 2016). Based on the result of single-factor design, a two-level Plackett–Burman design of 12 runs were inducted in this paper to confirm the important factors that significantly affect the extraction percent of mixed-acid. The standard ANOVA was calculated from experimental runs (Table 3). The analysis indicated A-hydrolysis temperature, D-TFA concentration and E-hydrolysis time had significant influence on extraction percent of purine. F-value of the model was 18.76 which indicated the model was significant. The value of p < 0.05 implied that the model terms were significant. Moreover, the predicted R2 (0.7596) was in reasonable agreement with the adjusted R2 (0.8898) as the different was <0.2. “Adeq precision” was used as a tool to measures the signal to noise ratio, that greater than four was desirable. In this study, a ratio of 13.549 implied an adequate signal and this model could be used to navigate the design space. As shown in Table 3, the contribution of hydrolysis time (38.76%) was the highest, which was followed by hydrolysis temperature (36.42%), TFA concentration (15.61%), FA concentration (3.04%), and liquid to solid ratio (0.16%). Moreover, as showed in Fig. 4, the variables of E, A, and D presented a positive effect on the total purine extract. By analyzing the data obtained from PBD, the insignificant variable was ignored and the variables of extraction temperature, time and the concentration of TFA were selected for further study by BBD to attain the optimal extraction process.

Figure 4 Pareto chart of Plackett–Burman design.

Table 3 The results of PBD in the screening experiment.

Score	Sum of squares	Contribution	df	Mean square	F-value	p-value		
Model	24.940		5	4.990	18.760	0.0013	significant	
A-Hydrolysis temperature	9.670	36.420	1	9.670	36.360	0.0009	significant	
B-Liquid-solid ratio	0.042	0.160	1	0.042	0.160	0.7047		
C-FA concentration	0.810	3.040	1	0.810	3.030	0.1323		
D-TFA concentration	4.140	15.610	1	4.140	15.580	0.0076	significant	
E-Hydrolysis time	10.290	38.760	1	10.290	38.690	0.0008	significant	
Residual	1.600		6	0.270				
Cor total	26.540		11					

Optimization of media using Box–Behnken design

PCA method

There were a total of 29 runs for optimizing the four independent parameters in the current BBD. Table S1 shows the experimental design used for the study. The data evaluated by ANOVA were presented in Table 4. These data were fitted to nonlinear quadratic models for percentage yield of total purine content. The fitted model equation is:Total purine yield(%)=85.85+0.47A+0.063B+0.27C−0.34D−0.085AB−0.52AC+          0.36AD+0.015BC−0.09BD+0.42CD−2.71A2−0.67B2−1.01C2+          0.001583D2

Table 4 Analysis of variance for the experimental results of the Box–Behnken design in PCA method.

Source	Sum of squares	df	Mean square	F-value	p-value		
Model	59.330	14	4.240	18.890	<0.0001	significant	
A-PCA concentration	2.660	1	2.660	11.860	0.0040	significant	
B-Hydrolysis temperature	0.048	1	0.048	0.210	0.6504		
C-Sample hydrolysis time	0.860	1	0.860	3.830	0.0707		
D-Liquie-solid ratio	1.410	1	1.410	6.300	0.0249	significant	
AB	0.029	1	0.029	0.130	0.7250		
AC	1.090	1	1.090	4.870	0.0446	significant	
AD	0.500	1	0.500	2.250	0.1561		
BC	0.0009	1	0.0009	0.004	0.9504		
BD	0.032	1	0.032	0.140	0.7096		
CD	0.710	1	0.710	3.140	0.0979		
A2	47.710	1	47.710	212.660	<0.0001	significant	
B2	2.920	1	2.920	13.010	0.0029	significant	
C2	6.610	1	6.610	29.470	<0.0001	significant	
D2	0.000016	1	0.000016	0.00007	0.9933		
Residual	3.140	14	0.220				
Lack of fit	2.610	10	0.260	1.980	0.2661	insignificant	
Pure error	0.530	4	0.130				
Cor total	62.470	28					

The Model F-value of 18.89 showed that this model was significant at less than 0.0001. The lack of fit F-value of 1.98 and the p-value of 0.2661 implied it was unimportant due to relative pure error. The not significant of value of lack of fit indicated the model was fitted with good prediction (Singh, Bajar & Bishnoi, 2017). In addition to, the value of the determination coefficient (R2), adjusted coefficient (Adj R2), and coefficient of variation were 0.9497%, 0.8994%, and 0.56%, respectively. The R2 value of the quadratic regression model showed that only 0.0503 of the total variations were not explained by the model. Adjust R2 of 0.8994, which also confirmed that the model was highly significant. Besides, the low value of the coefficient of variation clearly indicated a better precision and reliability in conducted experiments. The ANOVA indicated that the independent variables studied hydrolysis time, liquid-solid ratio, PCA concentration were significant factors (p-value <0.05) (Karacabey & Mazza, 2010), and the quadratic terms (AC-PCA concentration and hydrolysis time) significantly affected the total purine yield.

Contour plots were constructed to study interaction effects of the factors on the responses (Yin & Dang, 2008). Contour plots figures were drawn for the pair-wise combination of the four variables while keeping other factors at its medium value. Different shapes of contour plots indicated different type of interactions between the factors. The elliptical shape showed that interactions between corresponding factors were significant, although the reciprocal interactions between the corresponding factors are insignificant as represented by a circular contour plot (Kamble et al., 2018). The quadratic equation and ANOVA showed that AC (PCA concentration and hydrolysis time) have strength interaction. The interaction between PCA concentration and hydrolysis temperature was represented by contour plot shown in Fig. 5. As excepted, the contour plot was elliptical, which indicated the interaction among them was significant. Fig. 5 showed that PCA concentration exhibited a significant effect whereas hydrolysis time. An increase in total purine yield resulted when the concentration of PCA (A) was added in the range from 70 to 82 °C. Likewise, as the time of hydrolysis (C) increased in the range from 50 to 56, total purine extraction percent increased. Finally, the optimum conditional for PCA purine extraction method given by the software of Design-Expert 8.0.6 was acid concentration 80.97%; temperature 79.78 °C; hydrolysis time 54.20 min; liquid-solid ratio 50.78:1 and the extraction percent of total purine was 86.15. Then, according to the actual operating conditions, the optimum predicted process parameters were adjusted to acid concentration 80%; temperature 80 °C; hydrolysis time 55 min; liquid-solid ratio of 50:1 and the next verification test was carried out.

Figure 5 Contour plot of PCA concentration and hydrolysis time on extraction yield of total purine content.

Mix-acid method

As shown in Table 3, three most important variables affecting extraction efficiency of total purine were hydrolysis time, the temperature of hydrolysis and TFA concentration, according to the screening by Plackett–Burman design. These three factors were further optimized by BBD. The equations below show the relationship of the three factors and extraction yield.

Total purine yield(%)=104.26+0.39A+0.50B+0.35C+1.33AB−0.71AC−0.77BC−2.65A2          −0.72B2−1.24C2

The experimental design and ANOVA result of the model was shown in Table S2 and Table 5. The value of p less than 0.001 and non-significant value of lack of fit (F = 0.59, p-value = 0.6523) indicated the model is fitted with good prediction. The model F-value of 40.10 indicated the model was significant and only a 0.01% chance that “Model F-value” this large could occur due to noise. Moreover, the model has a high value of R2 (0.9810), Adj R2 (0.9565), and Pre R2 (0.8858), indicating a good agreement between the experimental results and predicted values of the response. Furthermore, in this paper “Adeq precision” of 17.524 > 4, which implied this model can be used to navigate the design space. At the same time, the p-values of A, B, C, AB, AC, BC, A2, B2, and C2 were all lower than 0.05 demonstrating that they were significant variables.

Table 5 Analysis of variance for the experimental results of the Box–Behnken design in the mix-acid method.

Source	Sum of squares	df	Mean square	F-value	p-value		
Model	57.05	9	6.34	40.10	<0.0001	significant	
A-Hydrolysis temperature	1.23	1	1.23	7.80	0.0268	significant	
B-TFA concentration	2.00	1	2.00	12.65	0.0093	significant	
C-Sample hydrolysis time	0.99	1	0.99	6.29	0.0405	significant	
AB	7.08	1	7.08	44.76	0.0003	significant	
AC	2.02	1	2.02	12.75	0.0091	significant	
BC	2.40	1	2.40	15.20	0.0059	significant	
A2	29.56	1	29.56	186.97	<0.0001	significant	
B2	2.21	1	2.21	13.98	0.0073	significant	
C2	6.47	1	6.47	40.92	0.0004	significant	
Residual	1.11	7	0.16				
Lack of fit	0.34	3	0.11	0.59	0.6523	insignificant	
Pure error	0.77	4	0.19				
Cor total	58.16	16					

The mutual effect of TFA concentration and hydrolysis temperature was represented in Fig. 6A. The counter plots show elliptical shaped indicating the significant interaction. Moreover, in a certain range, the extraction percent of total purine increased with TFA concentration and hydrolysis temperature. Increased TFA concentration and hydrolysis temperature led to the increase in total purine yield, but slightly decreased when exceeded the threshold levels of 89.00% and 92.00 °C. The interactions between hydrolysis time and temperature was significant because of elliptical shape of contour plot (Fig. 6B). An increase in the yield of total purine could be achieved with the controlled hydrolysis time was 8–13 min. Similar interactive effects were observed when TFA concentration and hydrolysis time were considered (Fig. 6C). It was obvious that the extraction percent of total purine was the highest when the hydrolysis time and TFA concentration were 10.20 min and 87.00%, respectively. Finally, BBD experiment predicted that the highest extraction percent was 104.51%, and the extraction conditions were as follows: mixture acid 88.00% TFA/80.00% FA (v/v, 1:1); temperature 91.83 °C; hydrolysis time 9.43 min; liquid-to-solid ratio 30:1. With some modifications, TFA concentration of 90.00%, temperature of 90.00 °C, hydrolysis time 10.00 min were used to verification test.

Figure 6 Contour plots of mix-acid method.

(A) Contour plot of TFA concentration and hydrolysis temperature on extraction yield. (B) Contour plot showing the effect of hydrolysis time and temperature on extraction yield of total purine. (C) Contour plot of TFA concentration and hydrolysis time on extraction yield of total purine.

Determination of the optimal method

In order to determine the best extraction method, the extraction yields of the two methods were compared; the results were given in Table 6. The actual extraction yield of total purine were 83.75% ± 2.65% (PCA method) and 103.94% ± 3.85% (mix-acid method), respectively. The relative error with the predicted value were less than 5%. The experimental results were well agreement with the predicted results. The result indicated that the model developed was considered to be accurate and reliable. In addition, the extractions of the total purine by the mixed-acid method were higher than those by the PCA method. These differences were likely due to protective effects of FA on purine bases. Moreover, TFA could promote complete dissolution of purine bases. In addition, the mixed-acid treatment was time saving and could effectively avoid the production of toxic chlorine. Therefore, the mixed-acid method was chosen for the extraction of purine bases from different marine fish.

Table 6 Optimum conditions, adjusted conditions, predicted and experimental value of response at that condition.

	Conditions	Total purine extraction yield (%)	
(a) validation test of PCA purine extraction method.	
	1A	2B	3C	4D	Predicted	Experimental	
Optimum	80.97	79.78	54.20	50.78:1	86.15		
Actual operating	80.00	80.00	55.00	50.00:1		83.75 ± 2.65	
(b) validation test of mix-acid purine extraction method.	
	aA	bB	cC	dD	eE	Predicted	Experimental	
Optimum	91.83	88.00	9.43	80.00	30:1	104.51		
Actual operating	90.00	90.00	10.00	80.00	30:1		103.94 ± 3.85	
Notes:

1 A-PCA concentration.

2 B-Hydrolysis temperature.

3 C-Hydrolysis time.

4 D-Liquid-solid ratio.

a A-Hydrolysis temperature.

b B-TFA concentration.

c C-The time of sample hydrolysis.

d D-FA concentration.

e E-Liquid-soild ratio.

Method validation

The linearity, range, and LOD of the method were subsequently calculated; results are shown in Table 7. Excellent linearity was observed for the quantification of the four purine bases. The LODs ranged from 0.0118 to 0.0774 mg/L, which were considered excellent. Furthermore, the R2 values were all above 0.9999, indicating a good linearity.

Table 7 Linear relation results (n = 3).

Purine	Regression equationa	R2	Linear range (mg/L)	LOD (mg/L)	
Adenine	y = 126196x − 16483	1.0000	0.1–300	0.0774	
Guanine	y = 90966x − 8366.6	1.0000	0.1–300	0.0178	
Hypoxanthine	y = 104998x − 8604.2	1.0000	0.1–300	0.0118	
Xanthine	y = 67689x − 1249.5	0.9999	0.1–300	0.0555	
Note:

a Variables: y, peak area (mV); x, concentration of each analyte (mg/L).

Standards were used to evaluate the precision of the method. For the repeatability and recovery tests, Scophthalmus maximus (dorsal muscles) samples were prepared by the method described previously. As shown in Table 8, the deviation in results were less than 0.69%, which indicated that the HPLC method was highly reproducible. The mean repeatability of the method for quantification of adenine, guanine, hypoxanthine, and xanthine were 1.32%, 0.87%, 0.83%, and 1.77%, respectively. By this method, average recoveries ranged from 94.90% to 104.51%. These results suggest that the mixed-acid method avoided damaging the purine bases, and confirm the validity of the method.

Table 8 Precision, repeatability, and recovery results (n = 6).

Purine	Precision (%)a	Repeatability (%)b	Recovery mean (%)c	
Adenine	0.13	1.32	95.31	
Guanine	0.69	0.87	94.90	
Hypoxanthine	0.02	0.83	104.51	
Xanthine	0.06	1.77	95.48	
Notes:

a Expressed as RSD by mixed purine base standards solution repeated six times.

b Expressed as RSD by Scophthalmus maximus (dorsal muscles) repeated six times.

c Average of recoveries at three spiked levels.

Purine contents in edible parts of marine fish

The contents of the four purines in different edible parts of marine fish were determined by the method described above; results are shown in Table 9. Among the marine fish evaluated, the total purine content (dorsal muscles + abdominal muscles + skin + entrails + eyes) was highest in Sphyraena. In contrast, Rajiformes exhibited the lowest total purine content. Considering only the edible parts of the fish (muscles, skin, and eyes) Scophthalmus maximus, Sphyraena, and Sardinella had the highest purine contents. Rajiformes and Anguillidae had the lowest purine contents and would thus be more suitable for a low-purine diet. To control the concentration of serum uric acid and avoid hyperuricemia and gout, it is best to reduce the consumption of high-purine fish such as Scophthalmus maximus and Sphyraena.

Table 9 Purine content in different parts of marine fish (mean ± S.D.).

Sample	Adenine	Guanine	Hypoxanthine	Xanthine	Totala	
Scophthalmus maximus (dorsal muscles)	127.50 ± 1.32	143.51 ± 4.52	820.71 ± 20.57	96.39 ± 5.33	1,188.12	
Scophthalmus maximus (abdominal muscles)	135.33 ± 2.74	155.33 ± 2.72	920.61 ± 2.63	91.55 ± 1.85	1,302.82	
Scophthalmus maximus (skin)	247.04 ± 4.57	1,190.03 ± 24.06	206.01 ± 5.55	42.65 ± 0.92	1,685.73	
Scophthalmus maximus (viscera)	378.05 ± 3.16	467.58 ± 13.23	256.01 ± 16.76	114.72 ± 8.87	1,216.36	
Scophthalmus maximus (eyes)	6.55 ± 0.00	3,820.77 ± 28.33	117.39 ± 0.53	–b	3,944.71	
Scomberomorus niphonius (dorsal muscles)	121.17 ± 0.34	135.86 ± 7.76	704.17 ± 23.14	22.32 ± 3.70	983.53	
Scomberomorus niphonius (abdominal muscles)	148.01 ± 2.42	177.65 ± 15.17	651.59 ± 7.80	17.66 ± 0.16	994.91	
Scomberomorus niphonius (skin)	194.95 ± 7.09	588.67 ± 26.14	314.62 ± 4.93	23.29 ± 1.34	1,121.53	
Scomberomorus niphonius (viscera)	326.38 ± 0.14	822.00 ± 9.37	1,015.87 ± 1.48	498.97 ± 0.44	2,663.22	
Scomberomorus niphonius (eyes)	140.63 ± 2.68	1,636.54 ± 69.73	255.58 ± 2.82	50.29 ± 0.80	2,083.04	
Pleuronectiformes (dorsal muscles)	110.85 ± 4.85	109.43 ± 3.42	503.69 ± 18.61	70.14 ± 2.12	794.11	
Pleuronectiformes (abdominal muscles)	157.77 ± 2.22	398.17 ± 1.10	645.52 ± 8.28	7.39 ± 0.41	1,208.85	
Pleuronectiformes (skin)	83.70 ± 4.77	234.05 ± 19.29	385.25 ± 9.18	2.31 ± 0.13	705.31	
Pleuronectiformes (viscera)	673.41 ± 8.00	980.73 ± 9.34	205.48 ± 0.56	223.56 ± 5.30	2,083.18	
Pleuronectiformes (eyes)	157.62 ± 0.56	1,390.48 ± 5.31	454.60 ± 10.66	3.69 ± 0.57	2,006.39	
Sea catfish (dorsal muscles)	107.96 ± 0.46	110.48 ± 0.13	525.52 ± 0.71	7.63 ± 0.01	751.59	
Sea catfish (abdominal muscles)	84.53 ± 0.78	101.70 ± 0.26	325.97 ± 1.09	6.99 ± 0.01	519.20	
Sea catfish (skin)	244.51 ± 3.73	463.46 ± 13.39	380.49 ± 0.07	48.56 ± 0.27	1,137.02	
Sea catfish (viscera)	496.76 ± 0.47	482.56 ± 0.09	305.94 ± 0.04	1.89 ± 0.01	1,287.15	
Sea catfish (eyes)	56.56 ± 0.01	807.51 ± 0.19	114.82 ± 0.05	31.73 ± 0.04	1,010.64	
Sardinella (dorsal muscles)	88.30 ± 0.05	62.15 ± 0.08	486.70 ± 0.80	37.47 ± 0.01	674.62	
Sardinella (abdominal muscles)	136.73 ± 0.15	106.22 ± 0.06	835.42 ± 0.78	52.04 ± 0.15	1,130.41	
Sardinella (skin)	118.90 ± 0.12	1,100.16 ± 0.35	918.66 ± 2.01	121.20 ± 0.05	2,258.91	
Sardinella (viscera)	429.03 ± 4.72	607.65 ± 2.39	549.02 ± 2.34	257.57 ± 0.46	1,843.26	
Sardinella (eyes)	69.94 ± 0.35	1,156.23 ± 0.52	161.82 ± 0.03	56.40 ± 0.05	1,444.39	
Sphyraena (dorsal muscles)	110.57 ± 0.24	84.06 ± 0.84	715.71 ± 0.49	8.85 ± 0.33	919.19	
Sphyraena (abdominal muscles)	101.74 ± 0.02	314.09 ± 0.70	656.65 ± 3.02	185.27 ± 0.10	1,257.75	
Sphyraena (skin)	130.45 ± 0.01	3,651.71 ± 0.95	567.29 ± 0.23	69.19 ± 0.26	4,418.64	
Sphyraena (viscera)	722.00 ± 2.58	952.23 ± 12.49	901.25 ± 0.33	144.24 ± 0.41	2,719.72	
Sphyraena (eyes)	51.70 ± 3.08	1,131.60 ± 0.46	51.47 ± 1.33	20.80 ± 0.00	1,255.57	
Anguillidae (dorsal muscles)	280.04 ± 0.01	94.15 ± 0.50	302.94 ± 0.52	8.95 ± 0.05	686.07	
Anguillidae (abdominal muscles)	186.96 ± 0.10	112.08 ± 0.00	227.65 ± 0.13	–	527.61	
Anguillidae (skin)	33.50 ± 0.12	173.94 ± 0.65	219.41 ± 3.01	59.71 ± 0.20	486.56	
Anguillidae (viscera)	761.48 ± 0.70	915.74 ± 1.99	275.33 ± 0.30	29.97 ± 0.07	1,982.53	
Anguillidae (eyes)	68.76 ± 0.02	1,252.41 ± 3.76	65.93 ± 0.08	19.08 ± 0.15	1,406.19	
Rajiformes (dorsal muscles)	89.50 ± 0.06	88.93 ± 0.16	372.90 ± 0.10	–	551.33	
Rajiformes (abdominal muscles)	74.51 ± 0.14	52.91 ± 0.32	249.52 ± 0.13	–	376.94	
Rajiformes (skin)	130.46 ± 0.03	514.27 ± 0.67	451.36 ± 0.14	13.89 ± 0.03	1,109.97	
Rajiformes (viscera)	277.50 ± 1.52	363.22 ± 0.37	349.68 ± 1.73	95.21 ± 0.15	1,085.61	
Rajiformes (eyes)	54.51 ± 0.06	704.83 ± 0.48	250.32 ± 0.81	16.10 ± 0.16	1,025.75	
Trichiurus lepturus (muscles)	134.50 ± 3.85	170.59 ± 0.77	878.57 ± 1.88	14.31 ± 3.84	1,197.97	
Trichiurus lepturus (viscera)	83.21 ± 0.31	736.40 ± 2.41	491.41 ± 1.70	166.93 ± 5.78	1,477.93	
Trichiurus lepturus (eyes)	609.51 ± 8.72	2,579.80 ± 60.13	418.67 ± 12.05	382.21 ± 2.44	3,990.19	
Notes:

a Total purine = Adenine + Guanine + Hypoxanthine + Xanthine.

b Not detected.

The purine contents in fish viscera were significantly higher than those in muscles. The total purine content in viscera ranged from 1,085.61 to 2,719.72 mg/kg, but the muscle samples contained only 551.33–1188.12 mg/kg (dorsal muscles) and 376.94–1302.82 mg/kg (abdominal muscles) total purines, with Sphyraena and Scophthalmus maximus exhibiting higher total purine contents than the muscles samples of other fish. This may be due to the higher visceral metabolic rate in the intestine of these species, which could promote the formation of purine bases. Furthermore, all muscle samples (dorsal muscles and abdominal muscles) were found to contain higher amounts of hypoxanthine than the other three purine bases. This observation is consistent with the report by Qu et al., (2017), who found that hypoxanthine content was the dominant purine in the muscles of Lateolabrax japonicus, followed by adenine, guanine, and xanthine.

The skins of the marine fish contained large amounts of guanine and hypoxanthine, which accounting for 74.22–95.48% of the total purine content. The total purine content of skin varied signiﬁcantly between the different species of marine fish. Anguillidae skin exhibited total purine content of 486.56 mg/kg, whereas the purine content of Sphyraena skin was 10 times higher. This might be related to the different habitats and growth patterns of the marine fish. Theoretically, the four purine bases can be transformed to uric acid in an equal manner, but hypoxanthine and adenine exhibit the greatest hyperuricemic effect (Clifford et al., 1976). The main difference between the purine contents in viscera compared with that of the eyes was the dominant purine. As seen in Table 9, viscera were rich in adenine, guanine, and hypoxanthine, whereas eyes contained the highest level of guanine. Furthermore, it is worth noting the purine content in fish eyes, which ranged from 1,010.64 to 3,990.19 mg/kg. In some regions, especially in China, consumers commonly eat fish eyes. Considering the high purine content of fish eyes and the pain and potential joint damage caused by hyperuricemia and gout, the consumption of fish eyes should be avoided, even though they are rich in nutrients such as collagen and unsaturated fatty acids.

Effect of boiling on purine content

Changes in the purine content of muscle during boiling

Previous studies have demonstrated that processes such as boiling, steaming, and microwaving can significantly reduce the purine content in foods. Among these processes, boiling is one of the most effective approaches to decreasing purine content. However, most studies on the effect of boiling on purine levels have mainly focused on meats such as chicken and beef, while less attention had been paid to the effect in aquatic organisms (Brulé, Sarwar & Savoie, 1989; Young, 1983). The changes in purine content during the boiling of marine fish measured during this study are shown in Fig. 7.

Figure 7 Effects of boiling on the purine contents in muscles.

(A) Dorsal muscles, (B) abdominal muscles.

Hypoxanthine was the major purine base measured in the muscle samples (Fig. 7). In the dorsal and abdominal muscles of Scophthalmus maximus, the amounts of adenine, guanine, and hypoxanthine decreased after boiling in water, but no changes were observed in the amount of xanthine present (Fig. 7). Among the purines measured, hypoxanthine was found to undergo the greatest reduction, from 867.90 to 258.24 mg/kg (70.24%) in dorsal muscles and from 933.94 to 442.98 mg/kg (52.57%) in abdominal muscles. Several studies have determined that hypoxanthine and free hypoxanthine-related compounds have good solubility and can be released easily from foods during cooking (Brulé, Sarwar & Savoie, 1989; Young, 1982). Our findings are also comparable to those of Lou, Lin & Benkmann (2001), who found that free purine bases in grass shrimp could be easily released and transferred to the cooking liquid during the cooking process. Furthermore, the contents of adenine and guanine decreased slightly after boiling; the amounts of adenine and guanine released into the cooking water were 35.89% and 15.92%, respectively, in dorsal muscles, and 32.21% and 29.18%, respectively, in abdominal muscles. The xanthine contents in muscles were relatively low to begin with, and exhibited nearly no change after boiling.

As for the effect of boiling time on changes in purine content, the contents of the four purine bases decreased more extensively with prolonged treatment time, and most purines in the dorsal muscles were reduced within 12–15 min. In comparison, the total purine and hypoxanthine contents in abdomen muscles decreased significantly within 0–3 min, declining by 64.63% and 49.13%, respectively (Fig. 7).

Changes in purine content of skin during boiling

Changes in the purine content of the skins were similar to those of muscles. Boiling reduced the amounts of purine base presents. As shown in Fig. 8, boiling led to a slight decrease in total purine content, from 1,596.68 to 1,117.41 mg/kg in Scophthalmus maximus skin. Hypoxanthine decreased rapidly (by 41.47%) in the first 3 min, after which time the rate of removal slowed down. Guanine was the major purine base in the skin samples; however, adenine and guanine decreased only slightly after boiling, declining by 35.71% and 23.20%, respectively. Boiling had the greatest effect on the removal of hypoxanthine among the four purines, with the content decreasing from 324.04 to 82.51 mg/kg.

Figure 8 Effect of boiling on the purine contents in skin.

Change in purine content of the cooking liquid during boiling

In order to explore the transfer of purine between the fish parts and the cooking water during boiling, the changes in purine content of the cooking water were also determined. As shown in Fig. 9, with increased boiling time, the contents of the four purines in the cooking liquid increased gradually over 15 min. The total purine content of the water increased remarkably within the first 3 min. The cooking liquid contained abundant levels of hypoxanthine, which was measured at concentrations approximately three times higher than those of the other three purines combined. The hypoxanthine content significantly increased within the first 9 min and remained stable thereafter. There was no significant change in xanthine content over 15 min of boiling. The adenine and guanine contents in the cooking liquid increased sharply from 3 to 15 min, by 161.99 and 180.10 mg/kg, to final contents of 200.26 and 230.94 mg/kg, respectively. These results confirm that the purines contained in the edible parts of the marine fish were transferred to the cooking liquid. This result agrees with the findings of Young (1982), who reported that the level of hypoxanthine in tissues decreased during cooking as it was removed by the cooking liquids.

Figure 9 Effect of boiling on the purine contents in boiling liquid.

Conclusions

The results indicated that marine fish contained high purine levels, with Scophthalmus maximus, Sphyraenus, and Sardine containing higher purine levels than the other fish tested. Moreover, the dominant purine and content of each purine varied signiﬁcantly between the different parts of the marine fish, where purine content in the viscera and eyes of the ﬁsh were higher than that in muscles. The muscle samples (dorsal and abdominal muscles) were found to contain higher amounts of hypoxanthine, whereas the major purine base in the eyes was guanine, the skin of the marine fish contained large amounts of guanine and hypoxanthine, and the viscera were rich in adenine, guanine, and hypoxanthine. We confirmed that boiling significantly reduced the purine contents in the fish, as the purines were transferred to the cooking liquid. Thus, boiling fish before eating could reduce the purine content, thereby reducing the incidence of hyperuricemia and gout.

Supplemental Information

Supplemental Information 1 The raw data of purine content in different parts of seawater were determined by HPLC.

Click here for additional data file.

Supplemental Information 2 The Box–Behnken experimental design of PCA method with four independent variables.

aTotal extraction yield (%) = Adenine extraction yield + Guanine extraction yield + Hypoxanthine extraction yield + Xanthine extraction yield

bExtraction yield (%) = (purine content in sample (mg)/sample mass (kg)) × 100

Click here for additional data file.

Supplemental Information 3 The Box–Behnken experimental design of mix-acid method with three independent variables.

aTotal extraction yield (%) = Adenine extraction yield + Guanine extraction yield + Hypoxanthine extraction yield + Xanthine extraction yield

bExtraction yield (%) = (purine content in sample (mg)/sample mass (kg)) × 100

Click here for additional data file.

We thank all people who contributed directly or indirectly to this work. We thank all people who contributed directly or indirectly to this work, We are grateful to National & Local Joint Engineering Research Center of Storage, Processing and Safety Control Technology for Fresh Agricultural and Aquatic Products-Bohai University for infrastructure this study.

Additional Information and Declarations

Competing Interests

Author Contributions

Data Availability

The authors declare that they have no competing interests.

Tingting Li conceived and designed the experiments, analyzed the data, contributed reagents/materials/analysis tools, authored or reviewed drafts of the paper, approved the final draft.

Likun Ren conceived and designed the experiments, performed the experiments, analyzed the data, prepared figures, and/or tables, authored or reviewed drafts of the paper.

Dangfeng Wang conceived and designed the experiments, performed the experiments, analyzed the data, prepared figures, and/or tables, authored or reviewed drafts of the paper.

Minjie Song performed the experiments, prepared figures, and/or tables.

Qiuying Li authored or reviewed drafts of the paper.

Jianrong Li conceived and designed the experiments, contributed reagents/materials/analysis tools, authored or reviewed drafts of the paper, approved the final draft, jianrong Li had overall responsibility for this project.

The following information was supplied regarding data availability:

The raw measurements are available in File S1.

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
