# Peer review of "Optimization of extraction conditions and determination of purine content in marine fish during boiling"

_PeerJ, doi:10.7717/peerj.6690_

## Round 0.1 · original submission · Major Revisions

While the reviewers find some merit in the paper, they raised serious issues in method optimization especially reviewer 2 and several more parameters need be presented to confirm method validness for quantification of purines which the authors need to present in their revised version

·

Basic reporting

-Some comments are highlighted in PDF file include text editing and reference details.

-Sample pretreatment (line 97) The authors should state that treatments have been done separately to each edible part of fish samples.

-Extraction rate on axes (figure 2 for example) is defined by (%). I think its is not rate, it is extraction percent since rates literally are related to time.

-In tables (1 and 2); Factors' levels should be presented in actual values instead of coded levels for clarity. also footnotes should be shortened and preferably clarified in text.

Experimental design

-Why 90 degree is used for temperature levels in PCA optimization (line 183) although it showed low extraction% in the prior screening (same for 50 degree level in mixed-acid method). I suggested the levels of this factor should be 70,75 and 80 degrees for PCA and start with 70 degree in mixed-acid to have equal intervals. Also the intermediate level of liquid-solid should be 70:1 (D2) in the same optimization to have a regular intervals between levels. In my opinion, as long as temperature does not greatly affect the extraction % compared to other factors in screening, it should not be included the the design so the number of samples in the design is efficiently reduced.
-Justification of five level design for mixed-acid procedure optimization should be provided.

Validity of the findings

-Is the used HPLC reported before. If yes, reference should be provided and no need for method validation. If the method not reported before, so the author should put some emphases on optimizing the chromatographic conditions especially changes in mobile phase composition as the separated four components (Figure 2) showed four fairly resolved peaks after a waste of running time (5 min). In this case a complete validation according to compendial guid lines should be provided plus a calculated system suitability parameters such as peak resolution, symmetry and column selectivity.

-Line 152, LOD should be calculated based on signal to noise ration or from calibration graph. The authors stated that "LOD was determined using standard solutions". The authors should provide more details about the used concentrations also justify the method used according to official reference.

-Line 156, to determine the % recovery of the method; actually there is nor % recovery for a method, it is for standard or sample solutions. Also how standards were added 0.5, 1 and 2 times the amount found in samples although samples concentration literally are unknown.

-Terms in equation (line 160) should be clarified. I see content added (presumably standard) is the denominator.. assuming that is the recovery for standard added not for the sample.

Additional comments

The manuscript discussed important and interesting topic for purine analysis in marine fish matrix. The manuscript is well written and clearly presented. However some comments are listed that may improve the quality of the work.

Reviewer 2 ·

Basic reporting

no comment

Experimental design

1. My major concern is regarding the optimization of the purines extraction methods, the authors choose to change one factor while keeping the other factors constant then used an orthogonal design to find the optimum parameters. This design does not allow the study of the interaction between the factors or the quadratic terms. Why the authors did not use a factorial design for screening such as Plackett-Burman designs. Then a central composite design for generating a statistically significant and optimum design that takes into account all the possible terms.
2. With regards to the HPLC method used for determination of purines, there was not any discussion of how this method was developed for example how the authors choose the mobile phase used during the separation process, what is the pka of the studied purines. Moreover, the authors did not even mention the final pH of the mobile phase which is a crucial parameter in the ionization and the separation of the studied compounds.

Validity of the findings

The authors should calculate the system suitability parameters regarding the HPLC method, different parameters should be calculated and addressed including, resolution, tailing factor, number of theoretical plates and retention factor. The authors may refer to the USP guidelines for the calculation of these parameters. Moreover, I see no baseline separation between Adenine and Guanine in Fig.1. Finally, the robustness of the method should be assesed

---

## Round 0.2 · accepted · Accept

The authors have satisfactory addressed reviewer comments

#